# Rapid and reversible photoinduced switching of a rotaxane crystal

Kai-Jen Chen[1], Ya-Ching Tsai[1], Yuji Suzaki[2], Kohtaro Osakada[2], Atsushi Miura[3] & Masaki Horie[1]

Crystalline phase transitions caused by external stimuli have been used to detect physical changes in the solid-state properties. This study presents the mechanical switching of crystals of ferrocene-containing rotaxane controlled by focused laser light. The expansion and contraction of the crystals can be driven by turning on and off laser light at 445 nm. The irradiation-induced expansion of the crystal involves elongation along the *a*, *b* and *c* axes at 30 °C, whereas heating of the crystal at 105 °C causes the shortening of *c* axis. The expansions reversibly occur and have the advantage of a rapid relaxation (reverse) process. Single-crystal X-ray crystallography reveals the detailed structural changes of the molecules, corresponding to a change in the size of the crystals on laser irradiation. This molecular crystal behaviour induced by laser irradiation, is demonstrated for the remote control of objects, namely, microparticle transport and microswitching in an electric circuit.

[1] Department of Chemical Engineering, Frontier Research Center on Fundamental and Applied Sciences of Matters, National Tsing Hua University, 101, Section 2, Kuang-Fu Road, Hsinchu 30013, Taiwan. [2] Chemical Resources Laboratory, Tokyo Institute of Technology, 4259 Nagatsuta, Midori-ku, Yokohama 226-8503, Japan. [3] Department of Chemistry, Faculty of Science, Graduate School of Chemical Sciences and Engineering, Hokkaido University, Kita-10, Nishi-8, Kita-ku, Sapporo 060-0810, Japan. Correspondence and requests for materials should be addressed to M.H. (email: mhorie@mx.nthu.edu.tw).

Phase transitions in the crystals of molecular compounds have been extended to spin-crossover transition-metal complexes[1,2], mixed valence compounds[3] and stimulus-responsive organic materials[4,5]. A recent application of crystalline phase transition involves positive or negative thermal expansion based on an anisotropic change of the crystal shape caused by a change in temperature[6–8]. Most previous studies have been concerned with full phase transition, where the structure, electronic state and/or conformation of all of the molecules within the crystals vary under crystallographic control[1,4,5,8–11]. Reversible processes for the repeated performance of crystals require a combination of a double stimulus for the forward and backward phase transition, that is, photoirradiation, temperature change, irradiation at different wavelengths and the backward process is generally 1–2 orders of magnitude slower than the forward process[12–16]. We reported the thermal phase transition of rotaxane crystals comprising ferrocenylmethyl (4-methylphenyl)ammonium and crown ether at 128 °C (heating) and at 115 °C (cooling)[17,18]. Photoinduced structural changes in the crystal would be a new path to a metastable crystal structure and provide a multiple stimulus–response system for a single compound. This study presents the rapid and highly reversible mechanical performance of a crystal phase transition caused by a local structural change using focused laser irradiation as the stimulus. In addition, the full phase transition occurs with a rapid, reversible process, resulting in a new metastable phase.

## Results

**Photoinduced structural changes.** Single crystals of rotaxane containing a ferrocenylmethyl(4-methypheynyl)ammonium axle molecule and a dibenzo[24]crown-8 (DB24C8) ring molecule

were used in this study. Figure 1 summarizes the observed changes in the molecular structures and interference colours. Focused irradiation with a 445 nm laser (12 mW, beam diameter ~1 μm, Supplementary Fig. 1) at 30 °C led to a rapid, slightly subtractive and interference colour, yielding a laser-induced (LI) phase accompanied by a crystal expansion (Fig. 1, path I). The crystals underwent rapid and reversible expansion/contraction when the laser beam (445 nm) was turned on/off at 30 °C. The length and width of the crystal top face increased from 86.1 to 86.3 μm (+0.2%) and from 46.1 to 46.6 μm (+1.0%) along the a and b crystal cell axes, respectively, and the thickness (c axis) increased from 15.4 to 15.5 μm (+0.6%) at the focal point of 12 mW laser irradiation. The crystal size reached a maximum size within 12 ms (Supplementary Fig. 2a). Turning off the laser resulted in contraction of the expanded crystal, returning it to the original size on the same timescale (12 ms) (Supplementary Fig. 2b). The crystal size was repeatedly changed for more than 1,000 times within 1,200 s by turning the laser light on and off (Supplementary Fig. 3). Thus, the high-speed switching of the mechanical motion (and change in the interference colour) of the crystal was achieved for both forward and backward paths by controlling the irradiation. The backward switching of this crystal occurs much faster than the bending motion of azobenzene groups reported for crystals (3.8 s) and polymer films (8 s–5 min)[15,19–21].

A photoinduced crystal-to-crystal phase transition occurred at 115 °C under 5 mW of focused laser irradiation (445 nm), resulting in the formation of a LI high-temperature (LI-HT) phase in 25 ms (Fig. 1, path II and Supplementary Fig. 2a). The interference colour of the resulting crystal appeared to be similar to that produced by a thermal crystalline phase transition at 128 °C reported earlier[17,18]. The LI-HT phase shows a change in the resulting crystal dimensions that are slightly larger than

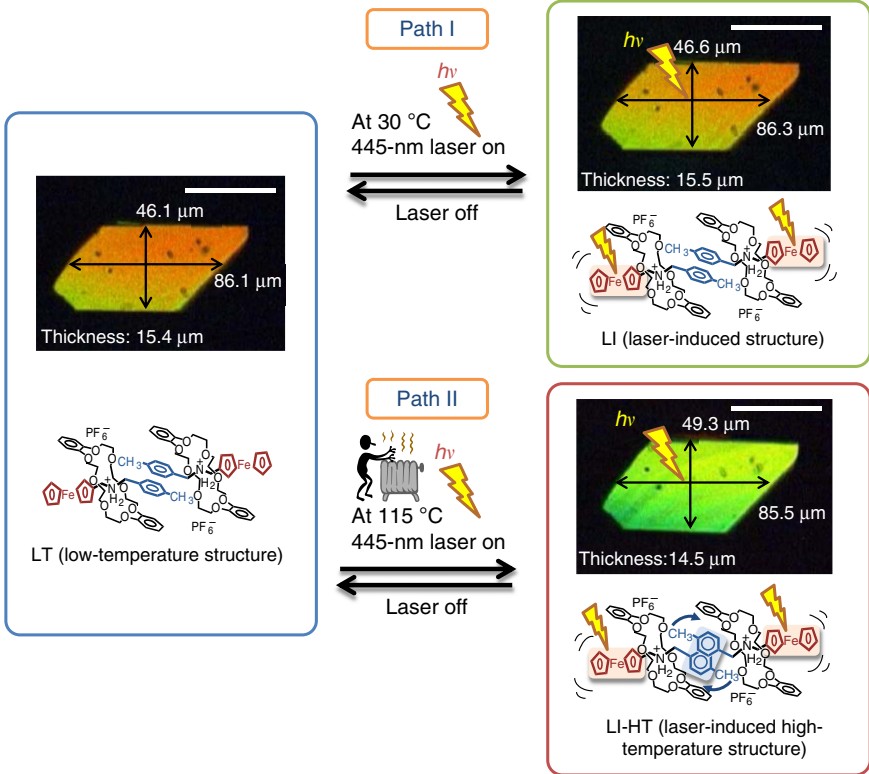

**Figure 1 | Optical micrographs of rotaxane crystal under polarized light.** Focused laser irradiation at 445 nm (12 mW) induces rapid and reversible expansion of the crystal from LT to LI at 30 °C (path I). Focused laser irradiation (5 mW) triggers rapid and reversible crystal-to-crystal phase transition from LT to LI-HT at 115 °C (path II). Scale bars, 50 μm. See also the Supplementary Movie.

the previously reported high-temperature (HT) phase crystal (width by $+0.5\%$, length by $+0.9\%$ and thickness by $+0.5\%$). The original crystal form was spontaneously recovered when the laser was turned off, and the backward transformation was completed within 25 ms at 115 °C (Supplementary Fig. 2b). In both experiments, the laser beam spot diameter was $\sim 1\,\mu m$, much smaller than the top face of the crystal ($86 \times 46\,\mu m^2$), suggesting that the local structural changes were rapidly propagated through the entire crystal via a co-operative domino effect motion[22,23] even under the thermal phase transition temperature (128 °C).

**Effect of laser irradiation and temperature**. The effect of laser irradiation on the surface temperature of the crystal was analysed through thermographic measurement (Fig. 2 and Supplementary Fig. 4). At 30 °C, the crystal temperature rapidly increased at the irradiated position from 30 to 40 °C by 445 nm wide-field laser irradiation (90 mW) (Fig. 2a). The laser irradiation (90 mW) at 115 °C increased the temperature of the irradiated position to 130 °C, and the temperature distribution in the crystal returned to its initial state immediately after turning off the irradiation (Fig. 2b). Thus, both crystal expansion at 30 °C and crystal-to-crystal phase transition at 115 °C are related to temperature increases induced by irradiation.

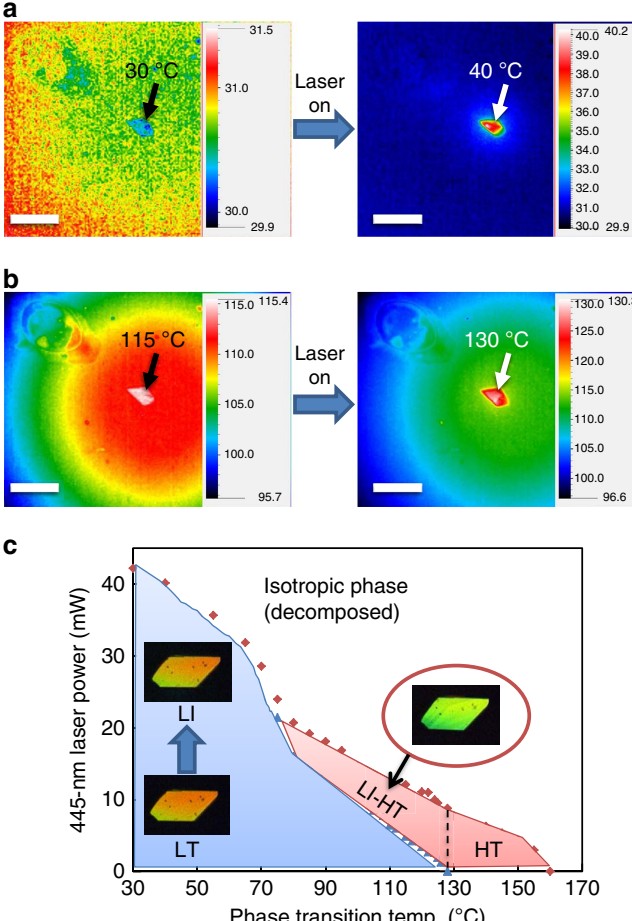

**Figure 2 | Effect of laser irradiation and temperature on the rotaxane crystal.** (**a**) Thermographic images of the crystal without and with 445 nm wide-field laser irradiation at 30 °C. (**b**) Thermographic images without and with laser irradiation at 115 °C. (**c**) Phase diagram of the crystal observed using a polarizing optical microscope at various temperatures and laser powers. Scale bars, 500 μm.

The phase diagram was plotted for a wide temperature range based on observations using a crossed polarized microscope (Fig. 2c). At temperatures below 80 °C, the crystal showed a reversible change from the low-temperature (LT) phase to the LI phase by 445 nm laser irradiation below 20–40 mW. At crystal temperatures between 80 and 128 °C, 445 nm 0.8–20 mW laser irradiation caused a change from the LT phase to the LI-HT phase. Above 128 °C, the HT phase was formed without irradiation. Thus, the phase of the rotaxane crystal can be controlled through combination of laser power and temperature.

The area of the (001) facet of the single crystal changes on irradiation, which varies depending on the power and wavelength of the laser at 30 °C. The magnitude of crystal expansion is also significantly dependent on the wavelength and power of the lasers (Supplementary Fig. 5a). A maximum expansion ($+0.9\%$) was observed using the 445 nm laser at 8.5 mW, while the same degree of change using a focused 1,064 nm laser required a power of 1.5 W. The rotaxane shows a maximum absorption wavelength at 440 nm, which is assigned to metal-to-ligand charge transfer in the ferrocenyl group (Supplementary Fig. 5b). Therefore, efficient crystal switching (expansion and mechanical motion) observed on irradiation at 445 nm is due to the photoexcitation of the ferrocenyl group, which then induces an increase in the crystal temperature and molecular movement. The molecular motion of the ferrocenyl group induced by the laser expands the thickness ($c$ axis) of the crystal.

The irradiation of single crystals of ferrocene with laser (445 nm) at 30 °C exhibited a minor degree of photoinduced deformation at a laser power of 2.5 mW and led to rapid decomposition and melting at a laser power of 4.5 mW (Supplementary Fig. 6). Thus, ferrocene behaves as a photosensitizer to convert light energy into mechanical motion and heat. The crystal of the ferrocene-containing rotaxane in this study undergoes photoexcitation of the ferrocenyl group, resulting in a change in the molecular structure of rotaxane and the formation of the LI phase that differs from the heat-induced (HI) phase obtained by thermal heating. Moreover, these molecular motions are amplified to change the shape and volume of the crystal on a macroscopic scale, which can be integrated into photocontrollable devices.

**Molecular structures**. The results of single-crystal X-ray crystallography under wide-field irradiation (445 nm at 30 °C) are presented in Fig. 3a–c. Each of the aromatic and ferrocenyl groups in the molecules are shifted towards the directions shown in Fig. 3a. The elongation of all axes $a$ ($+0.5\%$), $b$ ($+1.0\%$) and $c$ ($+0.2\%$) led to three-dimensional expansion of the crystal (Fig. 3b). This expansion corresponds to elongation of the distances between rings A and A′, B and B′, and A and Fe from 3.89 to 3.92 Å ($+0.8\%$), 10.60 to 10.70 Å ($+0.9\%$) and 8.04 to 8.06 Å ($+0.2\%$), respectively (Fig. 3c and Supplementary Figs 7 and 8). However, heating up to 105 °C without laser irradiation caused the elongation of the $a$ (0.6%) and $b$ (1.1%) axes and the contraction of the $c$ axis by 0.2% (Fig. 3d). The distances between rings A and A′ and B and B′ are elongated ($+1.2\%$ for both), while that between ring A and Fe is decreased to a small degree ($-0.1\%$) (Fig. 3e and Supplementary Figs 7 and 9). The elongation of the distance between ring A and Fe can be accounted for by the photoactivation of the ferrocenyl group in rotaxane, which causes the local motion of the molecule to elongate the $c$ axis despite the higher thermodynamic stability of the HI phase with a contracted $c$ axis relative to the LI phase.

Figure 3f compares the rotaxane molecular structure changes in the crystal on heating and laser irradiation. The LI phase is produced by laser irradiation at 30 °C and has elongated tolyl A–A′, catechol B–B′ and A–Fe distances, as well as expanded

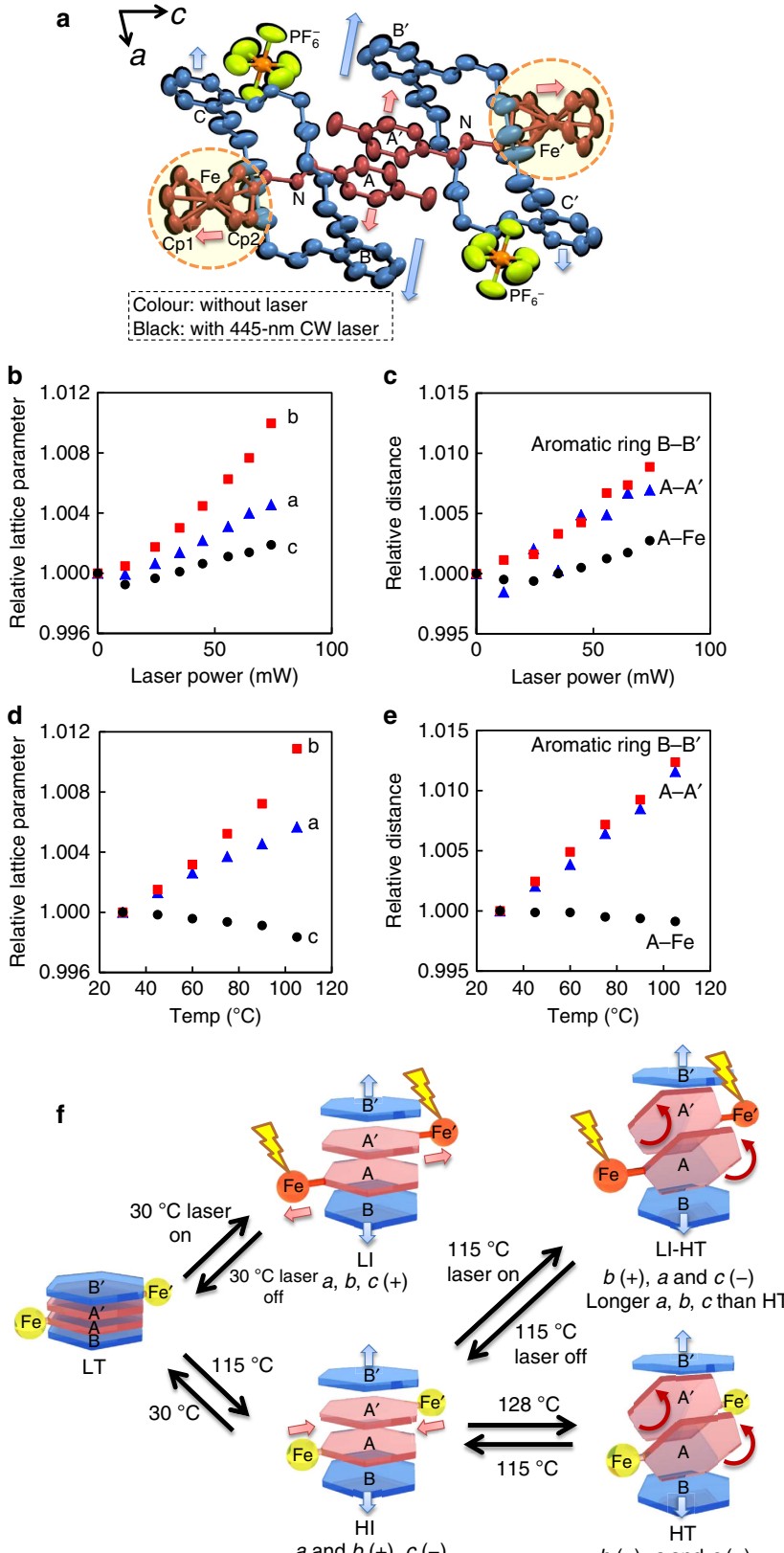

**Figure 3 | Molecular structures in crystal.** (**a**) Superimposed plots of molecular structures of LT at 30 °C (colour) and LI with 445 nm laser irradiation at 74 mW at 30 °C (black) with 50% probability. (**b**) The 445 nm wide-field laser power dependence of relative unit cell parameters *a*, *b* and *c*. (**c**) Relative inter-/intramolecular distance changes by 445 nm wide-field laser irradiation. (**d**) Temperature dependence of relative unit cell parameters *a*, *b* and *c*. (**e**) Relative inter-/intramolecular distance changes between 30 and 105 °C. (**f**) Schematic of photoinduced structural change in rotaxane. See also the Supplementary Movie.

crystal lattice in the *a*, *b* and *c* axes directions. Simple heating at 105 °C causes the formation of the HI phase with a shortened A–Fe distance and contraction of the *c* axis. Laser irradiation above 115 °C yields a crystal with the shape and interference colour nearly similar to those of the HT phase; however, crystals of the LI-HT phase have a slightly larger size in the *a*, *b* and *c* directions compared with the HT phase. The effect of the laser irradiation on the length of the *c* axis can be attributed to

the photoexcitation of the ferrocenyl group, resulting in a conformation change in the axle molecule similar to the change in the crystal caused by laser irradiation at 30 °C (from the LT phase to the LI phase).

**Microparticle transport and force**. The rapid and reversible size/shape change of the crystals described above can be readily applicable to the mechanical output in the visible scale range. Microparticle transport occurring due to the photoinduced mechanical crystal motion was demonstrated (Fig. 4a–c) using a silica microparticle (diameter 15–50 µm). The particle was deposited on the surface of a rotaxane crystal horizontally positioned on a glass substrate (Fig. 4a). When the laser (445 nm) was focused on the crystal immediately below the particle, the microparticle bounced off the crystal due to the increase in crystal thickness. When the crystal was vertically positioned and the particle was laterally placed, irradiation with the laser focused significantly close to the interface between the crystal and the attached particle, resulting in rapid translation of the particle in the vertical direction against the crystal (Fig. 4b). When the two particles were placed against both faces of the vertically aligned crystal (Fig. 4c), stepwise irradiation of a focused laser beam at the interfaces between the crystal and the particles selectively translated the particles one at a time. The forces accompanying the photoinduced crystal expansion and the phase transition were measured using a microforce analyser comprising two gauges on a cantilever (Supplementary Fig. 10). The photoinduced expansion of the crystal was reversible with a magnitude tuned by the laser power (Fig. 4d and Supplementary Fig. 11). The maximum weight under laser irradiation of the crystal was 68 mg at 17 mW (crystal size: $635 \times 565 \times 84\,\mu m$, weight: 41 µg), which is 1,650 times larger than the crystal weight. This weight ratio is similar to that observed for the phase transition and 1–2 orders of magnitude higher than that previously observed for photochromic azobenzene polymer films[24,25], diarylethene crystals[26,27] and salicylidenephenylethylamines crystals[28]. Such a facile single-wavelength process with a relatively quick response is highly advantageous. In addition, the force was detected along the vertical direction to the top surface of the crystal that corresponds to the *c* axis of the unit cell. Hence, such motion cannot be achieved directly by heating the crystal.

**Switching for electric circuit**. A microswitch in an electric circuit was demonstrated using a gold-coated rotaxane crystal (Fig. 5a). The rotaxane crystal was mounted on an iron needle and coated with a thin layer of gold (transmittance of ∼30% at 445 nm) by

**Figure 4 | Light-driven rotaxane crystal switching and microparticle transport.** (**a**) Schematic and photos of silica microparticle transport caused by the photoinduced mechanical motion of the rotaxane crystal. A 16 µm-diameter particle was placed on the surface of a crystal positioned horizontally on the glass substrate. Local laser irradiation (445 nm, 8 mW) at a position adjacent to the particle caused the jumping of the particle. (**b**) Translation of a particle placed on a vertically positioned crystal. The particle was deposited on the surface of a crystal that was vertically positioned on the glass substrate. Local laser irradiation (488 nm, 3 mW) at a position adjacent to the particle caused translation of the particle. (**c**) Translation of two particles. Two particles were placed against both faces of the vertically placed crystal, and the stepwise irradiation of a focused laser beam at the interfaces between the crystal and the particles selectively and separately translated the particles. (**d**) Time dependence of lifting weight of the crystal induced by 445 nm focused laser irradiation at various laser powers at 30 °C. Scale bars, 25 µm. See also the Supplementary Movie.

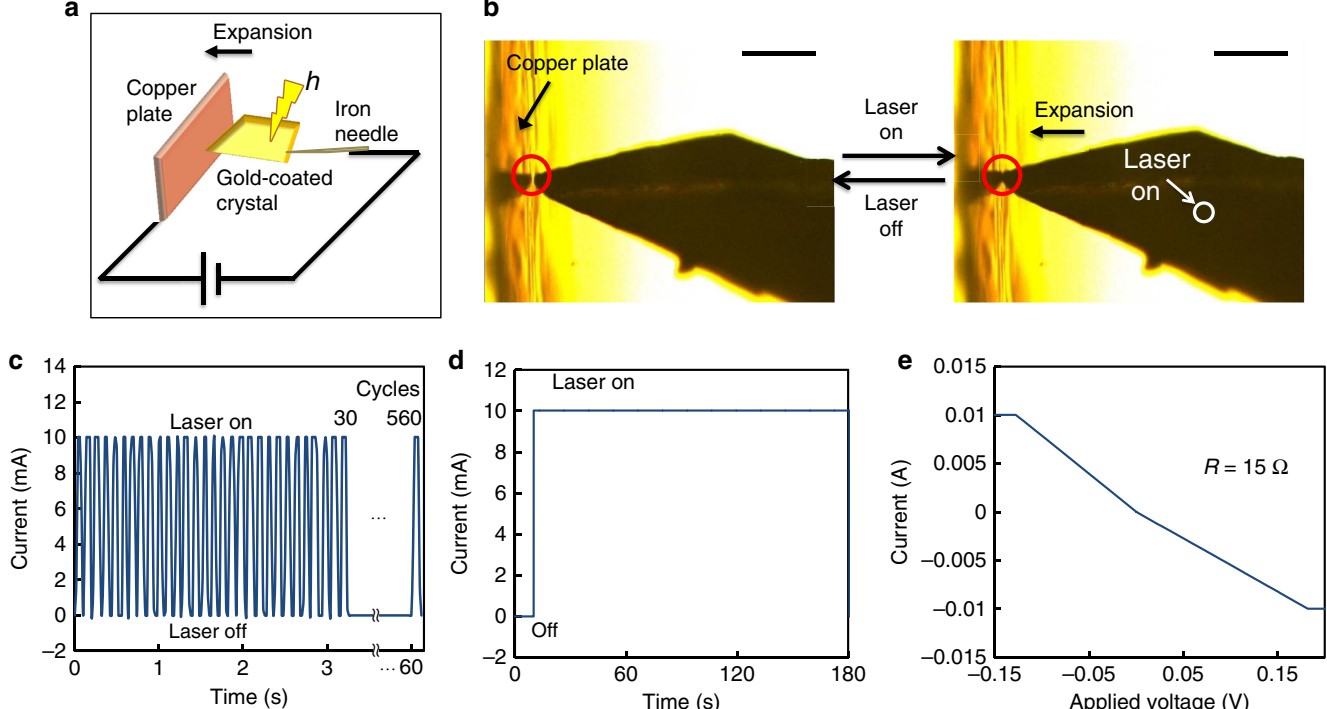

**Figure 5 | Light-driven rotaxane crystal switching for electric circuit.** (**a**) Schematic and (**b**) photos of the rotaxane crystal switch for the electric circuit. Gold-coated rotaxane crystal contacts on the copper plate due to 445 nm focused laser irradiation at 44 mW at 30 °C. Scale bars, 200 μm. (**c**) Electric current time dependence controlled by 445 nm laser irradiation at 44 mW. See also the Supplementary Movie. (**d**) Static measurement of electric current. (**e**) Dependence of electric current on applied voltage. See also the Supplementary Movie.

sputter deposition. The crystal was placed within a few micrometres of a copper plate, and these were connected to an electric circuit. Figure 5c shows the current changes induced by turning the laser switching on and off. Under irradiation, the crystal expanded and connected to the copper plate, rapidly increasing the current. After the laser was turned off, the crystal rapidly disconnected from the copper plate. This switching was repeated 560 times within 60 s. Moreover, the system is much faster and more facile than diarylethene photochromic crystals, which require two wavelengths to bend reversibly[29]. Furthermore, the electric circuit was extremely stable when the crystal was connected to the copper plate (Fig. 5d) with an electric resistance estimated to be 15 Ω (Fig. 5e).

## Discussion

The laser light-induced mechanical motions and phase transitions of ferrocene-containing rotaxane crystals were discovered under designed conditions. Owing to the presence of ferrocenyl groups, the rotaxane crystals rapidly convert light energy into three-dimensional mechanical motion through the expansion of the $a$, $b$ and $c$ axes derived from the elongation of the axle groups comprising ferrocene. In particular, the LI expansion along the $c$ axis, which corresponds to the thickness of the crystal, was found to be useful in photomechanical conversion for the microparticle transport. In addition, the rapid and reversible change in the crystal shape and size enabled the switching of an electric circuit. We anticipate that such photoinduced molecular motions will have more applications in molecular mechanical and optoelectric devices based on the simple integration of molecular building blocks.

## Methods

**General method.** Synthesis and crystallization of the rotaxane were performed on the basis of the method described in the literature[17,18]. The absorption spectrum was measured using a JASCO V-630 spectrophotometer.

**Photoinduced structure change measurement.** A continuous-wave 445 nm diode pump solid-state laser (TAN-YU, LSR445FP-1W) was used in this experiment. A multiple-optical-fibre cable was connected as an output light source and the light illuminated from the source was aligned by a SMA 905 collimator. After passing through a ×20 objective lens, the focused laser beam spot diameter was ∼1 μm. The power of the 445 nm laser beam passing through the objective lens was tuned by adjusting the amperage from the controller, and was measured using a power meter (OPHIR, Nova II P/N7Z01550). The crystals were observed using an optical microscope (Olympus, BX51) equipped with a ×20 objective lens with numerical aperture = 0.25. The Imagesource DFK 51AU02 CCD (charge-coupled device) camera with a video rate of 12 fps was used to capture the images presented in this article (Figs 1, 4 and 5 and Supplementary Fig. 6) and the crystal dimensions were measured by a confocal laser microscope with a ×50 objective lens (Keyence VK-9500). High-temperature experiments were performed using a hot stage (Linkam, HMS600) controlled by the corresponding controller (Linkam, TMS 92).

**Measurements of the time dependence of the displacement.** A complementary metal-oxide semiconductor (Lumernera, Infinity 1-3, CMOS) camera with an 80 fps video rate was used to capture the rapid crystal expansion motion (Supplementary Fig. 2). The expansion/contraction behaviour was observed under a ×50 objective lens (Olympus, LMPLFLN50X) with numerical aperture = 0.5, which enhances the imaging of the crystal displacement and increases the experimental precision.

**Thermographic measurement.** Infrared thermography accompanied with an infrared detector was used in this measurement (Ching Hsing Computer-Tech (CHCT), P384A, 20–23 μm). The thermos image has 384 × 288 pixels within the view area of 8.75 × 6.56 mm with the resolution of up to 22.78 μm per pixel, suitable for micro detection (the typical crystal size is 250 × 180 μm). This infrared detector has the frame rate of 50 Hz with the accuracy of ± 2%. The thermos images of the rotaxane crystal were recorded with various amounts of laser irradiation power.

**Single-crystal X-ray crystallography.** Single-crystal X-ray crystallography was performed for a rotaxane crystal at 30 °C (Bruker, APEX DUO, Dual Wavelength System, 60 Hz). An APEX II 4 K CCD Detector was used for data collection, and the corresponding Bruker APEX II software was used for data collection and reduction. The 445 nm laser fibre head was attached at the side of the crystal at a distance of 5 mm. The structures were solved using SHELXL-2014/6.

Crystal data and details of structure refinement are summarized in Supplementary Tables 1 and 2.

**Force measurement.** A microforce detector module (CHIEF SI, μ-force) and the corresponding Bridge DAQ software were used to measure the lifting weight of the rotaxane crystal induced by photoirradiation. This force detector module was composed of a cantilever attached to two strain gauges on both sides to enable the detection of small forces. During the measurement, all devices were placed on a vibration isolation optical table to prevent any possible vibration.

**Electric circuit measurement.** The rotaxane crystal was adhered on the tip of an iron needle, followed by deposition of a thin layer of gold by a Sputter coater (SPI MODULE) to provide the electric conductivity. The needle was then linked with the cathode of an electric circuit, and the anode with the rotaxane crystal was placed within a few micrometre of a copper plate. Both the anode and cathode sides of this circuit were connected to a series bipotentiostat (Digi-IVY: DY2300), which supplied 2 V of stable voltage during the measurements.

**Birefringence measurement.** Optical properties of single crystals were observed using a polarizing microscope (Olympus, BX51-P) with the combination of a temperature controller (Linkam, TMS92) and a hot stage (Linkam, THMS600) (Supplementary Fig. 12). The interference colours were converted into the optical anisotropic parameters of the optical retardations using the Michel Levy birefringence chart (Nikon interference colour chart) and a Berek compensator (Olympus, U-CBE) under observations with white light. The $\Delta n$ values were obtained from the optical retardation ($R$) and thickness of the crystal ($t$). The thicknesses of the crystals were measured using a confocal laser microscope with a $\times 50$ objective lens (Keyence, VK-9500).

**Data availability.** Data supporting the findings of this study are available within the article (and its Supplementary Information files) and from the corresponding author on reasonable request. Crystallographic data have been deposited at the CCDC and copies can be obtained on request, free of charge, by quoting the publication citation and the deposition numbers 1500708–1500721.

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

## Acknowledgements

We thank Pei-Lin Chen for single-crystal X-ray crystallography. This work was financially supported by Ministry of Science and Technology Taiwan.

## Author contributions

K.-J.C., Y.-C.T., Y.S. and M.H. carried out the X-ray measurements; K.-J.C., Y.-C.T., A.M. and M.H. carried out the laser experiments; K.-J.C., Y.S., K.O. and M.H. contributed to the editing of the manuscript.

## Additional information

**Competing financial interests:** The authors declare no competing financial interests.

**Publisher's note**: 

