## [Peer review file · Nature Communications]

Transferred manuscripts:

Reviewer #1 (Remarks to the Author):

This paper reports photostimulated mechanical response of a single crystal composed of ferrocene-containing rotaxane. In this study photoirradiation causes heating of the crystal. The difference between photoirradiation and normal heating procedure is the response of C-axis, Upon laser irradiation c-axis expands, while it contracts upon heating. The ferrocene units are the heat sources in the crystal. This may explain the expansion of the c-axis. Although the expansion by irradiation with laser is of quite interest, the photoresponse effect below 80° is rather small, less than 1%. Monotonous changes of the unit cell dimensions and the crystal volume indicate that the expansion does not include any phase transition. It is recommended to focus on the experimental results upon laser irradiation above 80°. At such high temperatures the phase transition takes place upon laser irradiation and the photoresponse effect is expected to become much larger.

Reviewer #2 (Remarks to the Author):

In my previous review I indicated that this work is interesting and important. However, I had concerns that that authors had not gone far enough in confirming that the changes are not simply due to heating as a result of laser irradiation. I believe that they have now at least presented a convincing case for light-induced changes due to electronic factors rather than just simple heating by irradiation. Since the authors have now included some discussion of heating effects I feel that the work, as is now stands, is suitable for publication in Nature Communications.

Reply to the comments by Reviewer #1 and points of amendment

Reviewer #1: This paper reports photostimulated mechanical response of a single crystal composed of ferrocene-containing rotaxane. In this study photoirradiation causes heating of the crystal. The difference between photoirradiation and normal heating procedure is the response of C-axis, Upon laser irradiation c-axis expands, while it contracts upon heating. The ferrocene units are the heat sources in the crystal. This may explain the expansion of the c-axis. Although the expansion by irradiation with laser is of quite interest, the photoresponse effect below 80° is rather small, less than 1%. Monotonous changes of the unit cell dimensions and the crystal volume indicate that the expansion does not include any phase transition. It is recommended to focus on the experimental results upon laser irradiation above 80°. At such high temperatures the phase transition takes place upon laser irradiation and the photoresponse effect is expected to become much larger.

Authors: We realized the importance of this point. We compared the force accompanying the crystal expansion and the photoinduced phase transition. In fact, these forces were observed to be very similar to each other. It might be due to the forward process for the crystal expansion (expansion with laser on), whereas the backward process for the phase transition (expansion with laser off). We realized 3 important advantages of the photoinduced expansion over the phase transition.

(1) Size: To carry out the photoinduced expansion, there is no limitation in crystal size. Any crystals size is applicable to mechanical switches. Contrary, small crystals (e.g. 50 x 50 x 20 μm^3) are preferable to carry out the phase transition because large crystals tend to be cracked during a repeating experiment because of too much strain in the crystals.

(2) Durability: As mentioned above, the crystals are much more durable with the photoinduced expansion than that with the phase transition. This is clearly good for further applications.

(3) Experimental setup: The photoinduced expansion requires the simple experimental setup using a laser only at a room temperature. On the other hand, the phase transition requires a temperature controller and a laser.

Consequently, to take up these advantages, we focused on the photoinduced expansion for the demonstrations in Figs 3 and 4. To briefly explain the force observed for the phase transition, we modified sentences as follows.

In page 7 line 13, "The forces accompanying the photoinduced crystal expansion and the phase transition were measured..."

In page 7 line 18, "This weight ratio is similar to that observed for the phase transition..."

Reply to the comments by Reviewer #2

Reviewer #2: In my previous review I indicated that this work is interesting and important. However, I had concerns that that authors had not gone far enough in confirming that the changes are not simply due to heating as a result of laser irradiation. I believe that they have now at least presented a convincing case for light-induced changes due to electronic factors rather than just simple heating by irradiation. Since the authors have now included some discussion of heating effects I feel that the work, as is

now stands, is suitable for publication in Nature Communications.

Authors: We sincerely appreciate your time and efforts to improve our manuscript. Thanks for your important comments in the previous submissions, we have obtained precious information of the photoinduced structural changes of the rotaxane crystals.

Reviewer #3 (Remarks to the Author):

Manuscript 101918 describes a photo-induced reversible single-crystal-to-single-crystal transformation of a ferrocene containing rotaxane. This assessment only focuses on the newly added crystal structures to the paper.

The crystallographic aspects of the paper are supported by well-behaved and converged structures. There are eight structures that correspond to the UV irradiation experiments and six related to thermal processes. The thermal structures were processed with SHELX-2014/6 while the set of UV structures only with SHELX97. The eight UV structures should be re-processed using the latest version of SHELX. This would ensure the cif file contains the instruction file used for refinement, hkl data, and more extensive CIF entries. Also, these structures processed with SHELX97 were found to be incomplete with a significant number of CIF cards not provided. Please make sure all CIF files are finalized. Once these are properly refined with SHELX2014, the crystallographic details provided in the paper should be in a suitable format for publication.

Reply to the comments by Reviewer #3 and points of amendment

Reviewer #3: Manuscript 101918 describes a photo-induced reversible single-crystal-to-single-crystal transformation of a ferrocene containing rotaxane. This assessment only focuses on the newly added crystal structures to the paper. The crystallographic aspects of the paper are supported by well-behaved and converged structures. There are eight structures that correspond to the UV irradiation experiments and six related to thermal processes. The thermal structures were processed with SHELX-2014/6 while the set of UV structures only with SHELX97. The eight UV structures should be re-processed using the latest version of SHELX. This would ensure the cif file contains the instruction file used for refinement, hkl data, and more extensive CIF entries. Also, these structures processed with SHELX97 were found to be incomplete with a significant number of CIF cards not provided. Please make sure all CIF files are finalized. Once these are properly refined with SHELX2014, the crystallographic details provided in the paper should be in a suitable format for publication.

Authors: The reviewer claims that photoinduced structures must be solved using the latest version of SHELX2014/6. This is an important point and we re-produced cif files accordingly. These new cif files are uploaded with this revised manuscript.